# Designer Functional Nanomedicine for Myocardial Repair by Regulating the Inflammatory Microenvironment

**DOI:** 10.3390/pharmaceutics14040758

**Published:** 2022-03-31

**Authors:** Chunping Liu, Zhijin Fan, Dongyue He, Huiqi Chen, Shihui Zhang, Sien Guo, Bojun Zheng, Huan Cen, Yunxuan Zhao, Hongxing Liu, Lei Wang

**Affiliations:** 1Guangdong-Hong Kong-Macau Joint Lab on Chinese Medicine and Immune Disease Research, The Second Affiliated Hospital of Guangzhou University of Chinese Medicine, Guangzhou 510120, China; chunping_liu@126.com (C.L.); markho1@126.com (D.H.); chenhuiqi0727@163.com (H.C.); huisozh1999@163.com (S.Z.); guosien66d@163.com (S.G.); zhengbojun@gzucm.edu.cn (B.Z.); danicacen@163.com (H.C.); eponinechaoi@163.com (Y.Z.); 2State Key Laboratory of Dampness Syndrome of Chinese Medicine, The Second Affiliated Hospital of Guangzhou University of Chinese Medicine, Guangzhou 510120, China; 3State Key Laboratory of Quality Research in Chinese Medicine, Institute of Chinese Medical Sciences, University of Macau, Macau 999078, China; 4Molecular Diagnosis and Treatment Center for Infectious Diseases, Dermatology Hospital, Southern Medical University, Guangzhou 510091, China; fanzhj5@mail.sysu.edu.cn; 5Department of Urology, Guangzhou Institute of Urology, Guangdong Key Laboratory of Urology, The First Affiliated Hospital of Guangzhou Medical University, Guangzhou Medical University, Guangzhou 510230, China

**Keywords:** myocardial repair, nanomedicine, inflammatory microenvironment, molecular imaging

## Abstract

Acute myocardial infarction is a major global health problem, and the repair of damaged myocardium is still a major challenge. Myocardial injury triggers an inflammatory response: immune cells infiltrate into the myocardium while activating myofibroblasts and vascular endothelial cells, promoting tissue repair and scar formation. Fragments released by cardiomyocytes become endogenous “danger signals”, which are recognized by cardiac pattern recognition receptors, activate resident cardiac immune cells, release thrombin factors and inflammatory mediators, and trigger severe inflammatory responses. Inflammatory signaling plays an important role in the dilation and fibrosis remodeling of the infarcted heart, and is a key event driving the pathogenesis of post-infarct heart failure. At present, there is no effective way to reverse the inflammatory microenvironment in injured myocardium, so it is urgent to find new therapeutic and diagnostic strategies. Nanomedicine, the application of nanoparticles for the prevention, treatment, and imaging of disease, has produced a number of promising applications. This review discusses the treatment and challenges of myocardial injury and describes the advantages of functional nanoparticles in regulating the myocardial inflammatory microenvironment and overcoming side effects. In addition, the role of inflammatory signals in regulating the repair and remodeling of infarcted hearts is discussed, and specific therapeutic targets are identified to provide new therapeutic ideas for the treatment of myocardial injury.

## 1. Introduction

Acute myocardial infarction (AMI) has become a major global health problem and a difficult barrier in the cardiovascular field due to its low five-year survival rate and high morbidity and readmission rates [1]. Causes of myocardial injury include ischemia, dilation, chemotherapy-related, sepsis, diabetes, etc. [2], which can be divided into ischemic and non-ischemic. Acute ischemic injury is the most characteristic cause of myocardial injury. Coronary artery occlusion after acute plaque rupture can lead to coagulation necrosis, programmed apoptosis, and secondary apoptosis. After myocardial injury, cardiomyocytes lose their envelope integrity, and the released cell fragments become endogenous “danger signals”, which are recognized and bound by cardiac pattern recognition receptors (PRRs) [2]. Initiating microbe pathogen-associated molecular patterns (PAMPs) or damage-associated molecular patterns (DAMPs) activate resident immune cells of the heart [3], release thrombin factor and inflammatory mediators, activate fibrinogen and platelets, and form thrombi. These effects all lead to myocardial cell contraction, enhanced permeability, microcirculation perfusion disorders, and hypoxia, leading to severe inflammatory reactions [2,4,5].

The mechanism of heart failure after AMI is closely related to the process of ventricular remodeling [6], and the degree of reconstruction depends on the size of the infarction focus and the quality of cardiac repair [3]. Inflammatory signals play an important role in dilation and fibrosis remodeling of infarcted hearts and are key events driving post-infarct heart failure [7,8]. The traditional treatment methods for myocardial injury include antibiotics, antioxidants, and surgery, but the purpose of clinical treatment is only to delay the progression of myocardial injury due to a lack of more effective treatment measures. Currently, the diagnostic methods of cardiac injury include cardiac color Doppler ultrasound, computed tomography (CT), cardiac magnetic resonance imaging (MRI), and cardiac radionuclide imaging. However, the specificity of these methods needs to be improved, and misdiagnosis or missed diagnosis may occur occasionally. Therefore, it is of great clinical significance and urgency to find new diagnostic and treatment strategies.

In view of the serious threat of cardiovascular disease to human health, myocardial injury should be widely considered and emphasized by researchers, and interdisciplinary integration of various fields should be encouraged to promote the development of new methods in the diagnosis and treatment of cardiovascular disease. The development of cardiovascular drugs has become a top priority in the field of new drug development. According to Evaluate Med Tech’s calculations, the cardiovascular drug market is now more mature in the international market, with global sales expected to be close to $100 billion, making it the second largest drug class after oncology drugs [2].

Nanomedicine, a novel scientific field in which nanoparticles are utilized in the prevention, treatment, and imaging of diseases, has spawned a series of beneficial applications, thanks to the rapid development of materials science, nanotechnology, and biomedical engineering. “Nanoparticles” is the abbreviation of nanoscale particles, which refers to materials with at least one dimension in three-dimensional space at nanometer size (1–100 nm) or with nanometer-sized substances as the basic unit. Nanoparticles have a high specific surface area and can be modified by various ligands to give them different biological functions, allowing for the design of a variety of targeted nanomedicines. In general, the design of nanodrug delivery systems can solubilize drugs, improve their half-life, improve drug distribution in vivo, enhance targeting, and reduce toxicity and side effects, which has great application potential in the field of disease diagnosis and treatment [9,10]. Therefore, the application of nanomedicine in the diagnosis and treatment of injured myocardium has important research significance and application prospects. This descriptive review discusses the therapeutic approaches and challenges of myocardial injury and describes the advantages of functional nanoparticles in improving therapeutic effectiveness and overcoming side effects. In addition, this review discusses the role of inflammatory signals in regulating the repair and remodeling of ischemic myocardium and attempts to identify specific therapeutic targets, providing new ideas for the diagnosis and treatment of injured myocardium.

## 2. Status of Cardiovascular Inflammatory Diseases

The diagnosis of myocardial injury inflammatory lesions is a difficult and challenging task, and despite various imaging modalities, myocardial injury inflammatory lesions are still a diagnosis of exclusion [11]. Currently, endomyocardial biopsy (EMB) combined with genomics and immunohistochemistry can be used to evaluate myocardial inflammatory lesions in clinical practice. However, due to its invasiveness, sampling variability, limited spatial information, and low clinical implementation, it is difficult to apply in clinical practice and has great limitations [12,13]. For inflammatory lesions of myocardium injury, clinical treatment strategies are supportive treatment or treatment of the primary underlying diseases, which are focused on delaying the disease process, are unable to reverse and repair the myocardial injury, and have limited effects on reducing morbidity and mortality. Therefore, it is of clinical urgency to actively seek effective treatment methods for myocardial injury and inflammation.

### 2.1. Existing Methods of Diagnosis for Myocardial Injury Inflammatory Lesions

Most of the early stages of cardiomyopathy are subclinical, that is, without obvious symptoms [14]. As the disease progresses, patients develop symptoms of cardiac insufficiency. The patient’s personal history, such as chemotherapy, history of diabetes, major trauma, etc., and clinical symptoms are the main basis of diagnosis of cardiomyopathy. Laboratory examination and radiographic analysis of myocardial injury caused by abnormal heart function or structure are essential. Evaluation methods include electrocardiogram, echocardiography, coronary computed tomographic angiography (CTA) and cardiac magnetic resonance imaging (MRI), etc. The patient’s medical history, symptoms and signs, and examination results are analyzed to exclude specific cardiomyopathy and endemic cardiomyopathy, and a diagnosis is finally reached [12]. However, the above diagnostic methods have great limitations in accuracy and sensitivity for early myocardial injury [15], which is one of the main causes of missed diagnosis and misdiagnosis in clinical practice. Therefore, timely and accurate assessment of the myocardial inflammatory microenvironment can help in the timely clinical diagnosis of myocardial injury.

### 2.2. Current Treatment Methods for Myocardial Injury and Inflammatory Lesions

Clinical treatment of various myocardial injuries and inflammatory lesions mainly follows two principles. One is to provide symptomatic support for the hemodynamic disorders of heart disease itself, including conventional anti-heart failure, anti-arrhythmia, anticoagulant thrombolytic, inhibition of cardiac remodeling, improvement of myocardial metabolism, etc., to prevent further damage to the cardiac cells and maintain cardiac function and hemodynamic stability. The other is targeted management for the primary underlying diseases that induce myocardial injury [16,17,18], such as septic cardiomyopathy. According to international guidelines for management of sepsis and septic shock, anti-infective therapy should be immediately given after diagnosis to treat primary sepsis [19,20,21]. For diabetic cardiomyopathy, reasonable control of blood glucose, improvement of systemic and tissue insulin sensitivity, myocardial glucose uptake, and cardiac function are key factors to reduce the incidence and mortality of heart failure [22,23,24].

When myocardial injury occurs, the homeostatic function of the cell population is unbalanced, and the stressed cells release pro-inflammatory factors, chemokines, exosomes, and senescence-associated secretory phenotypes (SASPs), leading to the formation of a pro-inflammatory microenvironment and myocardial tissue dysfunction, exacerbating the process of myocardial injury [25,26,27,28,29,30,31]. With the development of cardiovascular precision medicine, targeted drugs, and the proposal of new therapeutic concepts to regulate the microenvironment of myocardial inflammation, it is possible to reverse and repair myocardial damage.

### 2.3. Problems That Need to Be Solved

The inflammatory response is an important link in the pathogenesis and development of coronary heart disease, and the value of inflammatory signals in the prediction and prognosis evaluation of acute coronary syndrome of coronary heart disease, as well as the anti-inflammatory effects of drug therapy, is increasingly being recognized [32]. There are several potential approaches for the diagnosis and treatment of cardiac inflammatory diseases, and targeted nanomedicine is one of the most promising approaches (Figure 1).

#### 2.3.1. Hydrogel

Hydrogels with particle sizes between 1–1000 nm can be used for drug delivery, medical diagnosis, as biosensors, and for biological separation [33]. Hydrogels are similar to natural soft tissue and can provide a microenvironment for the extracellular matrix, showing great potential in tissue engineering applications. In particular, the development of biodegradable conductive hydrogels is of great significance for repairing electrically active tissues such as the myocardium, skeletal muscle and nerves [33,34,35]. In recent years, hydrogels have been widely used in the diagnosis and treatment of myocardial injury and inflammatory lesions.

Zhao Chao et al. [36] invented a temperature-sensitive and biodegradable poly(lactic acid)-poly(ethylene glycol)-poly(N-iso-propylacrylamide) (PLLA-PEG-PNIPAM) block polymer, which could form nanofiber gel microspheres (NF-GMS) and be loaded with cardiocytes (CM) for cardiac regeneration and repair. This research can be applied to cardiac myocyte transplantation, significantly improving cardiac function and opening up broad application prospects in the regeneration and repair of the heart. Ghanta et al. [37] used a small molecule modified alginate to encapsulate rat mesenchymal stem cells (MSCs)in a core-shell hydrogel capsule, and constructed TMTD-alginate MSC capsules which improve ventricular function and remodeling in a post-MI rat model. Li Hekai et al. [38] developed a mixed alloy nanoparticle (AuNP)-hyaluronic acid (HA) hydrogel matrix for encapsulating human induced pluripotent stem cell-derived cardiomyocytes(iPS-CM) to overcome this limitation. MMP-2 reactive hydrogel was prepared by crosslinking with degradable peptides of matrix metalloproteinase-2 (MMP-2) using methacrylate-modified HA as the skeleton. RGD peptide was introduced as an adhesion point to enhance its biocompatibility. The addition of AuNPs regulates the mechanical and topological properties of the matrix by significantly increasing its stiffness and surface roughness, thus accelerating the formation of gap junctions in iPS-CM and coordinating calcium processing through the αnβ1 integrin-mediated ILK-1/P-Akt/GATA4 pathway. Transplanted AUNP-HA-hydrogel-encapsulated iPS-CM formed stronger gap junctions in infarcted mouse hearts and resynchronized ventricular electrical conduction after myocardial infarction. Hydrogel-delivered iPS-CMs play a strong role in angiogenesis, which also contributes to the recovery process.

#### 2.3.2. Nano-Enzyme

As powerful biocatalysts, enzymes have high specificity and catalytic efficiency in in vivo and in vitro biochemical reactions under relatively mild conditions. However, due to their disadvantages, such as high preparation and purification costs, poor operation stability, sensitivity to the reaction environment, and difficulty in recycling, nano-enzyme technology has emerged as an alternative [39]. As a kind of artificial enzyme, nano-enzymes have many unique advantages, such as low cost, high stability, easy modification, and adjustable catalytic activity. They can solve the limitations of natural enzymes and have been widely studied and applied [40].

Nano-enzymes have the functions of protecting the myocardium and improving Alzheimer’s disease and ischemic stroke, and their application research has been extended from in vitro to in vivo, providing new ideas and methods for the diagnosis and treatment of myocardial injury and inflammatory lesions [41]. Zhang Yue [42] et al., with metal nanoparticles as the enzyme activity center, constructed a hybrid nanozyme which not only has SOD- and CAT-like activities, but can also overcome the biological barrier targeting cardiomyocyte mitochondria, providing a new treatment for alleviating myocardial injury.

Nitric oxide (NO) is an important signaling molecule in the cardiovascular system and is associated with the pathogenesis of ischemic cardiomyopathy, septic cardiomyopathy, and other cardiomyopathies. One of the effective treatment strategies for these cardiomyopathies is to provide a controlled and constant supply of nitric oxide (NO). Li Haiyun et al. [43] constructed nitric oxide synthase (NOS-like NanoNOS), comprised of a noble metal nanoparticle core and mesoporous silica shell, and proved the catalytic capacity of NanoNOS for NO production. NanoNOS inhibits endothelial cell adhesion factor expression by releasing NO and protects endothelial cells from stimulus-induced damage and the resulting mono-endothelial cell adhesion, suggesting that NanoNOS therapy can help prevent myocardial injury. This study broadens the biomedical application of nanomases and provides a new idea for the prevention and treatment of diseases related to myocardial injury.

#### 2.3.3. Extracellular Vesicles

Extracellular vesicles (EVs) are lipid bilayer particles without the ability to replicate that are secreted by cells into the extracellular microenvironment [44]. Exosomes have a wide range of manifestations in terms of size, source, biochemical components, and biological functions, including intercellular communication, immune response regulation, and disease progression [45], and can be divided into four subgroups: exosomes, microvesicles, apoptotic bodies, and cancer bodies, among which exosomes are the most well studied. Exosomes have been found to be secreted by endothelial cells, cardiac progenitor cells, cardiac fibroblasts, and cardiomyocytes, suggesting that exosomes play a potentially important role in cardiovascular diseases [46]. Recent advances in extracellular vesicle research have not only highlighted their importance in cardiac physiology and pathology, but also attracted attention since these extracellular vesicles may have utility in the diagnosis and treatment of cardiac inflammatory diseases [47], particularly those that are functionalized by nucleic acids [48].

Anselmo Achille et al. [49] identified CD172a+ EVs as cardiogenic EVs, and found that cardiomyocytes (CMs) shed more EVs under hypoxic stress, thus promoting the positive inotropic effect of unstressed CMs. Clinically, patients with aortic stenosis with higher levels of CD172a+ EVs in the circulatory center have a higher survival rate than those with lower levels of CD172a+ EVs during transcranial aortic valve replacement. These results suggest that CD172a+ EVs are a promising prognostic biomarker for myocardial diseases.

Cardiac progenitor cell-derived exosomes (CPCS-EXs) are of cardiac protection and repair value. Li Xin et al. [50] found that exosomes secreted by cardiac progenitor cells can protect cardiac myocytes in viral myocarditis models. Its protective function is achieved by reducing virus proliferation and regulating the mTOR, Bcl-2, and Caspase signaling pathways. This study identified the anti-apoptotic effect of CPCS-EX in Coxsackie virus B3-infected cells and rats, suggesting that CPCS-EX may be an effective tool for the treatment of viral myocarditis, and this work provides a potential cell therapy for diseases such as viral myocarditis.

#### 2.3.4. Other Nanomaterials

Nanotechnology, due to its unique physical and chemical properties, presents many attractive prospects in the field of cardiovascular diseases [51]. The potential of drug delivery technology based on lipid nanoparticles has been fully demonstrated. For example, nano-probes are mainly used for targeted imaging of biochemical changes in life systems [52]. The combination of drug delivery integrating with diagnosis and therapeutic monitoring is an interesting evolution of this concept, which will contribute to the development of general targeted drug delivery tools for the effective treatment of cardiovascular disease and some other diseases [53]. Zhao Xueli [54] assembled 17β-estradiol nano-probes with peptides that can specifically bind primary cardiomyocytes and used perfluorocarbons as the core to obtain a new nano-probe, PCM-E2/PFPs. Both in vivo and in vitro studies have demonstrated that these PCM-E2/PFPs can be used as contrast agents when exposed to low-intensity focused ultrasound (LIFU). The significantly accelerated release of 7β-estradiol enhances the efficacy of the drug without systemic side effects. PCM-E2/PFPs and low-intensity focused ultrasound therapy also significantly increased the myocardial targeting and circulation time. In-depth treatment evaluation showed that PCM-E2/PFPs and low-intensity focused ultrasound significantly inhibited myocardial hypertrophy, especially with lower expression levels of β-MHC, Collagen 1, and Collagen 3 among other treatment groups, revealing the high efficacy and protective effect of myocardium targeted delivery. This new integrated nano-platform can provide a diagnostic and treatment carrier for myocardial injury diseases and has very broad application prospects.

Creatine kinase is a commonly used marker of myocardial injury, and its activity is an important indicator of myocardial injury [55]. Li Xingjun et al. [56] synthesized monodisperse spherical EU-QPTCA rare earth MOF nanomaterials with a controllable particle size by the ligand exchange method, and used oleic acid and acetic acid as modulators. EU-QPTCA nanomaterials show strong red luminescence. Studies have found that both ATP and ADP can quench the red luminescence of EU-QPTCA nanomaterials, and ATP has a stronger quenching effect than ADP. Creatine kinase can catalyze the formation of ADP and phosphocreatine from ATP and creatine at pH 9.0, thus restoring the fluorescence of EU-QPTCA nanomaterials. Therefore, this nanomaterial can be used as a novel fluorescent probe for the highly sensitive and specific detection of creatine kinase activity, providing a new approach to the detection of creatine kinase activity in human blood, and has important significance for the early diagnosis of myocardial injury.

## 3. Application of Nanomedicine Targeted to Cardiovascular Inflammatory Diseases

Myocardial tissue is mainly comprised of cardiac myocytes and surrounding stromal cells, including macrophages, fibrocytes, dendritic cells, mast cells, and small numbers of B lymphocytes and T lymphocytes, which together constitute the myocardial microenvironment. When the myocardium is damaged, the myocardial cells and stromal cells trigger a complex inflammatory cascade, releasing a large number of pro-inflammatory factors, chemokines, and reactive oxygen species, which together constitute the myocardial inflammatory microenvironment.

Due to their high specific surface area, nanoparticles can achieve efficient drug loading and targeted modification, which has attracted extensive attention in the field of cardiovascular targeted drug therapy. In recent years, a large number of functional nanoparticles have been designed to target specific cells and regulate the myocardial inflammatory microenvironment. This section mainly summarizes the application progress of nanoparticles in targeting myocardial injury and stromal cells and regulating the vascular injury inflammatory microenvironment (Table 1).

### 3.1. Nanomedicine Modulates the Microenvironment during Cardiovascular Inflammatory Diseases

Nanotechnology, because of its unique physical and chemical properties, presents many attractive prospects in the field of cardiovascular diseases. Combined with the advantages of nanodrugs in the diagnosis and treatment of diseases, good research results have been achieved in the treatment of myocardial inflammatory diseases by nano-drugs [57,58,59].

Myocardial metabolism produces reactive oxygen species (ROS), and normal myocardial tissue contains many antioxidant substances that can remove or assist in the elimination of ROS, thus protecting cardiomyocytes from ROS damage [60,61,62]. Studies have shown that in viral myocarditis, effective ROS scavengers can resist ROS damage to myocardial cells and reduce apoptosis, providing new ideas for the treatment of viral myocarditis [67]. Liu Chao et al. [57] prepared TTPCD NPs, active oxygen scavenging nano-drugs targeting mitochondria, and then prepared ATTPCD NPs by simultaneously encapsulating the anti-inflammatory polypeptide AC2-26, which has anactive oxygen scavenging ability and anti-inflammatory effect. Such a noninvasive atomized nano-drug provides a new approach to the prevention and treatment of cardiac inflammatory diseases and other acute/chronic heart diseases related to oxidative stress. Dai Chengli et al. [58] proposed a biosafe *ganoderma lucidum spore oil* nano-system to enhance the anti-radiation and protective heart function of traditional spore oil, which is a safe and long-term agent protective against radiation, and studied its anti-radiation activity and related action mechanism as well as its heart-protective function.

When myocardial injury occurs, macrophages transform into the M1 subtype, triggering inflammation, while the M2 type is conducive to tissue regeneration [66,67,68,69]. Tokutome M. et al. [59] formulated poly (lactic acid/glycolic acid) nanoparticles containing pioglitazone (pioglitazone-NPs), which could promote macrophage polarization toward a pro-healing M2 phenotype and attenuate cardiac remodeling in an MI model without reperfusion. In addition, they also found that CCR2-dependent inflammatory monocyte recruitment is the primary target of pioglitazone-NPs. Xue Y. et al. [60] prepared macrophage membrane coated nanoparticles (MMNPs) containing miR199a-3p (MMNPmiR199a-3p). The MMNPmiR199a-3p inhibited the inflammatory response in MI mice and converted a M2 phenotype environment in peri-infarct zone. It is worth noting that the cardiovascular cells HL-1 cells only took up NPs containing Cy5-labeled miR199a-3p but not the free Cy5-labled miR199a-3p, which indicates the barrier of this delivery approach. Zhou Jin et al. [61] developed a melanin nanoparticle (MNP)/sodium alginate (Alg) hydrogel, which can achieve MI treatment based on the combination of reactive oxygen species (ROS) scavenging to suppress the oxidative stress damage and macrophage polarization to regenerative M2 phenotype in the MI microenvironment. Chen Jingli et al. [62] established a supramolecular nano-system to solve the problems of an excessive inflammatory response and early diagnostic delay in the MI process, which controlled targeted and phosphatidylserine (PS)-targeted drug delivery through an external magnetic field, enhanced drug accumulation in the infarct area, and accelerated the resolution of the early inflammatory response. The nano-system showed good inflammation resolution and imaging ability in MI model rats, achieving accurate visualization operation and significantly improving the early treatment effect of MI.

Neutrophils also play a key role in myocardial inflammatory diseases [68]. During myocardial ischemia, the activation of the complement system and chemokine production are triggered, which can cause aggregation of neutrophils in and around small blood vessels and capillaries around the ischemic area. When the ischemic myocardium regains an oxygen supply and reperfusion, the whole ischemic area accumulates neutrophils, aggravating myocardial injury [69,70,71]. Richart Adele L [63] et al. found that a single intravenous injection of N-apolipoprotein AI (apolipoprotein AI nanoparticles) immediately after myocardial infarction in a mouse model reduced the expression of chemokines that induce chemotaxis of neutrophils and monocytes to the heart. It also reduced the expression of monocyte surface integrin CD11b and reduced inflammatory responses in the heart and throughout the body. These data highlight the anti-inflammatory effects of N-apolipoprotein AI and provide preclinical support for its use in percutaneous coronary interventions to treat acute coronary syndromes. Molly M. Stevens et al. [64] reported a neutrophil-mediated delivery system that delivers drug nano-carriers to inflammatory tissues by utilizing the inherent ability of neutrophils to migrate to inflammatory tissues. This strategy shows the potential of nanocarrier-loaded neutrophils as a universal platform for delivering anti-inflammatory drugs that promote tissue regeneration in inflammatory diseases.

### 3.2. Nanomedicine Boost Immunotherapy for Cardiovascular Inflammatory Diseases

Myocardial injury promotes the inflammatory response, which manifests as activation of innate immune cells, tissue repair/remodeling, adaptive immune cell activation, and angiogenesis/fibrosis [72,73,74,75]. One of the first immune cell types to respond to injury is tissue-resident macrophages. Macrophages are essential for myocardial development, including the normal formation of the cardiac lymphatic system, and they promote appropriate electrical conduction [76,77,78]. Macrophages have recently been found to play an interesting role in maintaining myocardial energy homeostasis, mediated by processing depleted/defective mitochondria excreted by cardiomyocytes and engulfed by resident macrophages. Depletion of myocardial macrophages disrupts this process and leads to inflammasome activation and cardiac insufficiency [79,80,81].

T cells, outside of their role in host defense and their metabolic adaptability play, a crucial role in homeostasis [82]. T-cell metabolic failure leads to the accumulation of circulating cytokines, which is similar to the chronic inflammatory characteristics of aging [83], and metabolic disorders can cause myocardial inflammatory responses related to aging and the phenotypic characteristics of aging [84,85,86]. M2-like macrophages, natural killer cells, and Tregs are particularly promising targets for therapeutic intervention [87]. Other engineered immune cells, such as CAR macrophages, may also be on the horizon [88], and medical nano-carriers are one of the powerful tools to achieve immune cell modification.

Nanomedicine has undergone extensive modification of immune cells and has been successful in delivering drugs to damaged cardiac tissue through noninvasive delivery methods such as intravenous injection [89]. Qiaozi et al. [65] constructed a bionic hierarchical nano-targeted delivery system of neutrophils, which is made up of a hybrid membrane shell (artificial lipid membranes modified with FH peptide and fused with neutrophil mem- brane proteins (NMPs)), as well as a mesoporous silica nanoparticle core loaded with miRCombo (MSNs-miR). It can precisely deliver four micro-rRNAs to cardiac fibroblasts to achieve in vivo, in situ reprogramming. The results showed that the hierarchical targeting system could effectively deliver the four micro-RNAs to cardiac fibroblasts both in vitro and in vivo without affecting their role in promoting cardiac reprogramming. The histology and echocardiography results showed that the nano-targeted system could achieve direct cardiac reprogramming in vivo, reduce local fibrosis, and improve cardiac function after myocardial injury.

Teng Su et al. [66] prepared dual targeting biomimetic nano-cells by utilizing the natural targeting properties of bioactive compounds and platelets and combining them with biomimetic nano-cells. The team modified the platelet membrane with prostaglandin E2 and then wrapped the platelet membrane around the nanoscale bionic cells, resulting in bionic nano-cells with dual targeting. It was verified that the biomimetic nano-cells have good targeting ability in the injured heart and can significantly improve the cardiac function of the injured heart in animal models. This biomimetic nano-cell has dual targeting and therapeutic effects on the damaged heart region: on the one hand, it targets the damaged blood vessels with the help of platelet membranes on the surface, and at the same time, it uses prostaglandin E2 to bind to the receptors on the surface of the damaged cardiac cells, delivering therapeutic protein growth factor to the damaged cardiac tissue with precision. These results of functional nanomaterial targeting are expected to provide a new therapeutic strategy for the targeted repair of myocardial injury.

## 4. Application of Functional Nanomaterials in the Visualization of Cardiovascular Injury

Imaging technology allows noninvasive cardiovascular testing, disease staging, and monitoring of response to treatment, providing a powerful tool for the diagnosis and treatment of cardiovascular injury. Traditional imaging techniques include CT, PET, ultrasound, and optical imaging to obtain anatomical definitions and functional information, such as vascular reactivity and myocardial perfusion, activity, stiffness, and contractility, by identifying changes in the physical characteristics of lesions. Since it is difficult to fully understand the disease process at the molecular level, the pathophysiological characteristics of the disease cannot be directly revealed, which greatly reduces the information and accuracy of diagnosis. The development of molecular imaging techniques has provided an approach beyond anatomy for the diagnosis of cardiovascular disease by observing in vivo processes related to myocardial injury, such as inflammation, angiogenesis, apoptosis, oxidative stress, and fibrosis. Molecular imaging technology relies on the affinity or interaction between the contrast agent and target and has excellent specificity. However, the traditional small molecule contrast agent has a single function and rapid metabolism, which makes it difficult to meet the complex clinical needs in specific use.

The versatility of nanoparticles (NPs) provides a basis for increasing the long cycle, improving the targeting and biocompatibility, and improving the signal intensity of the unit target, which is expected to make up for the deficiency of traditional contrast agents. NPs have excellent radio-tagging efficiency and rapid synthesis and purification strategies, which are highly desirable for nuclear imaging applications using short-lived radioisotopes. In addition, nanoparticles with very high specific surface areas can achieve efficient loading of contrast agents and targeted modification, which has attracted extensive attention in the field of molecular imaging. In recent years, a large number of functional nanoparticles have been designed to visualize the inflammatory microenvironment during cardiovascular injury. This section will mainly summarize the application progress of nanoparticles in nuclear imaging, optical imaging, ultrasound imaging and multimodal imaging to visualize the inflammatory microenvironment of cardiovascular injury (Figure 2).

### 4.1. Nuclear Imaging

The nuclear imaging technique is a new technique combining nuclear technology and modern image theory. These methods include X-ray tomography (XCT), positron tomography (PET), nuclear magnetic resonance computed tomography (NMR–CT) and single photon emission tomography (SPECT). The common principle of the various nuclear imaging technologies is to obtain detailed internal information of the object to be studied by using the attenuation law or distribution characteristics of physical quantities related to nuclear radiation in the object to be measured, then use a computer to process this information at a high speed, and finally reconstruct an image of the object to be studied. Among them, PET, SPECT, and MRI are the most advantageous imaging methods in cardiovascular imaging. Because they can play roles in functional imaging, they are expected to achieve early detection before the occurrence of organ lesions.

PET and SPECT are 3D tomography techniques with high sensitivity for nanomolar or even picomolar detection. They are widely used as noninvasive clinical imaging tools. However, they also have limitations, such as low spatial resolution (1–4 mm) and ionizing radiation risk. A further challenge is that the requirements of a cyclotron for radioisotope production and its short half-life (typically a few hours) may reduce the time available to perform the required quality control. In addition, the commonly used contrast agents, such as 18FDG, may be absorbed and create a large amount of background due to metabolically active myocardium during PET detection of inflammatory myocardial injury rich in macrophages, which reduces the specificity and sensitivity of detection.

Multifunctional contrast agents based on nanoparticles provide a new strategy for detecting and diagnosing cardiovascular injury. Nanoparticles have unique sizes and physicochemical properties and can be loaded with a variety of radioactive tracers using different synthesis strategies. One of the main advantages of radioactive nano contrast agents is the presence of a large number of radioactive atoms in a uniform nanoparticle. The unique surface chemistry of nanoparticles also allows for improved targeting of disease sites using a variety of probes. This improves the contrast of the disease site relative to other normal tissues and improves the sensitivity and specificity of PET. Ueno et al. [90] used dextran nanoparticles doped with the isotope Cu-64 to track the immune-regulatory effects of angiotensin-converting enzyme inhibitor therapy on allograft bone marrow cells. The experimental results showed that the PET signal of nanoparticles in allografts was much higher than that of the background. The target-background ratio was also increased in allograft recipients, reflecting lower uptake of nanoparticles by non-inflammatory hearts. Keliher et al. [91] reported that a modified poly-glucose nanoparticle could achieve PET imaging of ischemic heart disease due to its high affinity for macrophages. The nanoparticles were enriched in cardiac macrophages to increase PET signaling in infarcted myocardium in mice. In the field of PET imaging, the most thoroughly studied nanomaterials include liposome carriers, magnetic ions, quantum dots (QDs), polymerized nanoparticles, silica nanoparticles, dendritic macromolecules, inorganic metallized nano-formulations, and carbon nanotubes [92].

MRI is a noninvasive clinical diagnostic tool that utilizes nonionizing radio waves. With high spatial resolution (~100 μm) and unlimited tissue penetration, MRI is capable of simultaneously imaging cardiovascular anatomy, physics, and molecular events. Contrast enhancers, such as gadolinium (Gd^3+^)-based drugs, are commonly used to enhance MRI imaging. Magnetic nanoparticles (MNPs) have lower toxicity and better contrast performance than gadolinium MRI imaging agents, and are playing an increasingly important role in imaging cardiovascular injury. MNPs can exude into the infarcted myocardium and aggregate in the target area, thus achieving angiography effects and greatly improving the early diagnosis of acute myocardial infarction.

Recent studies have found that MNPs targeting inflammation have a better detection effect. Park et al. [93] found that MNPs had a better detection effect than gadolinium-enhanced and manganese-enhanced MR in acute myocardial infarction models and could provide information on the inflammatory response in AMI mouse models. Cuadrado et al. [94] reported that magnetic nanoparticles targeting EMMPRIN could achieve visualization of acute myocardial infarction. In this study, EMMPRIN was targeted using paramagnetic/fluorescent micellar nanoparticles conjugated with EMMPRIN binding peptide AP-9 (NAP9). Cardiac magnetic resonance (CMR) scans revealed enhanced signaling in the left ventricle of mice injected with NAP9 compared to mice not injected with NAP9. These results suggest that magnetic nanosystems have potential applications in myocardial injury, such as myocardial infarction.

### 4.2. Optical Imaging

Optical imaging is widely used in preclinical studies for cell, subcellular and whole animal imaging due to its low cost and high detection sensitivity (picomolar level). Fluorescence imaging is one of the most commonly used optical imaging techniques, in which light of an appropriate wavelength is used to excite endogenous or externally introduced fluorescent parts and to detect light emitted at a longer wavelength. However, the effects of light scattering and tissue attenuation limit the application of visible fluorophores in deep tissue imaging. The fluorophore activity in the near infrared region can reduce the tissue attenuation effect and increase the penetration depth up to several centimeters, showing a certain prospect for in vivo detection.

Lu [95] et al. developed photodynamic selenium nanoparticles (SeNPs) targeting inflammatory macrophages. They coated selenium nanoparticles with chitosan (CS) via Se-S bonds, combining RB-CS-GSH with the photosensitizers Bengal red Rose (RB) and glutathione (GSH). The RB-CS-GSH layer was combined with catalase for imaging and photodynamic therapy. Second, they conjugated the carboxyl groups of hyaluronic acid (HA) and folic acid (FA) with the amine groups of ethylenediamine (EDA) to form an HA-EDA-FA mixture as the second layer of the nanoparticles. HA and FA are used as binding ligands of folate receptor β (FR-β) and CD44 on the surface of inflammatory macrophages. In vitro studies have shown that SeNPs provide a stronger RB fluorescence signal in lipopolysaccharide (LPS)-stimulated macrophages than in nonstimulated macrophages. This suggests that SeNPs can specifically bind to active macrophages via folic acid and HA coating. Inactive macrophages are not damaged by these nanoparticles. The results show that these nanoparticles have potential therapeutic and imaging applications.

Kosuge et al. [96] designed a single-walled carbon nanotube (SWNT) functionalized with Cy5.5 dye for near infrared (NIR) imaging and photothermal ablation of inflammatory macrophages. In vitro NIR imaging showed a strong signal after ligation of the left carotid artery in SWNT-treated mice, but no NIR signal in non-SWNT-treated mice. In general, single-walled carbon nanotubes have been shown to be therapeutic nanomaterials capable of fluorescence imaging and photothermal ablation of vascular macrophages. Ziegler et al. [97] constructed ROS-responsive self-assembled fluorescent nanoprobes for imaging and treatment of cardiac ischemia–reperfusion injury. In this study, reactive fluorescent ROS nanoprobes were evaluated in a mouse model of myocardial infarction and found to be highly specific for the ischemic/reperfusion myocardium in the first 24 h after reperfusion. Sun [98] et al. developed a novel nanoparticle based on three-function viroid anthropoid virus 40 (SV40) to deliver hirudin. The nano-system is also loaded with near-infrared quantum dots and labeled with a cyclic peptide (CGNKRTRGC) to target the P32 protein on macrophages. In ApoE^−/−^ mice, the fluorescence signal of the targeted nanoparticles in the plaque was three times stronger than that of the nontargeted nanoparticles. However, it is limited by fluorescence quenching and photobleaching. For deep cardiovascular tissue, fluorescence imaging is also limited by its penetrating ability.

Photoacoustic imaging (PAI) is a new optical imaging technology that uses light as an excitation source to generate thermal plucking by using the photo-thermal effect of target molecules, and it generates light absorption images by detecting the transient acoustic signals generated by the target through ultrasonic transducers. It provides high spatial resolution (50–500 μm) and penetration depth of information up to 5 cm. Both intrinsic tissue molecules (such as hemoglobin, lipids, water, and melanin) and exogenous photo-thermal agents (such as ICG, GNPs, etc.) can be used as contrast agents for imaging, and multi-wavelength imaging can be realized due to the differences in their absorption peaks. Therefore, photoacoustic imaging has gained wide attention in cardiovascular diseases.

Qin et al. [99] demonstrated for the first time that a class of photoacoustic nanoparticles (PANPs) containing semiconductor polymers (SPs) can be used as contrast agents for photoacoustic imaging (PAI) of HesC-CMS transplanted from living mouse hearts. In a recent study, Zhang et al. [100] used photoacoustic imaging of myocardial infarction regions in a rat myocardial ischemia–reperfusion model with noninvasive fibrin-targeted nanoparticles. This work provides a new idea for the detection of myocardial injury and the diagnosis of myocardial infarction by noninvasive real-time imaging.

In another interesting study, Zhao et al. [101] synthesized ultrasmall NPs via single-stranded DNA (ssDNA)/metal ion complex self-assembly. Animal experiments showed that the ultrasmall NPs could significantly enhance the PA signal intensity in the area of myocardial infarction. Gifani et al. [102] recently constructed super-selective nanoparticles targeting LY-6Chi monocytes and foam macrophages, and combined them with clinically feasible photoacoustic imaging (PAI) to accurately and specifically image inflammatory plaques in vivo in a mouse model simulating human vulnerable plaques. In conclusion, photoacoustic imaging has good application prospects in the field of cardiovascular injury and is expected to be translated into clinical practice.

### 4.3. Contrast Ultrasound Imaging

Molecular ultrasound imaging offers the possibility of real-time, noninvasive visualization of molecular markers of cardiovascular disease using clinical ultrasound imaging systems. The technique uses particles or nanoparticles to target binding to functional specific epitopes after systemic injection. These bound particles (contrast agents) are acoustically active in the presence of ultrasound, resulting in backscattered signals from the molecular events of interest, which can be located and imaged by a two-dimensional ultrasonic scanning system. Contrast agent ultrasound imaging of inflammation has been achieved by targeting contrast agents to activated immune cells or to endothelial cell adhesion molecules that regulate leukocyte trafficking and adhesion. Myocardial contrast echocardiography is used for tissue perfusion imaging, extending the concept of ultrasonic red blood cell tracking to visualization of intracardial microcirculation blood volume. During echocardiographic imaging, microvascular transport of microbubbles leads to a transient increase in myocardial tissue video intensity, which can be quantified and regionally mapped to myocardial perfusion.

This method has been applied to the detection of coronary artery disease and the evaluation of myocardial activity after acute myocardial infarction. Several commercial microbubble formulations for perfusion imaging are currently being evaluated in multicenter clinical trials. The application of microbubble molecular imaging takes advantage of these properties, namely, its intravascular position and acoustic responsiveness, to specifically detect molecular signatures on the endothelial surface. Unlike free-circulating microbubbles used for ventricular sedation and perfusion imaging, targeted microbubbles adhere to endothelial cells through specific ligand–receptor interactions pre-specified in the microbubble design. Targeted ligands attach to the surface of micro-vesicles, causing them to adhere to specific endothelial markers. This adhesion is shown in two-dimensional ultrasound images as an increase in the video intensity of the molecular target location region, and the adhesion persists even after the circulating microbubbles are washed out.

In recent years, ultrasonic microbubble contrast agents have been rapidly developed, and a wide variety of components have been reported, often including perfluorocarbons or nitrogen in shells made of phospholipids, albumin, or biodegradable polymers. Microbubbles are small enough to remain in the vessel, so the molecular target must be luminal. Other ultrasound contrast agents for non-gas-encapsulated microspheres include liposomes and liquid perfluorocarbon emulsion nanoparticles, which can exit the intravascular space and have the potential to acoustically identify extravascular targets, such as atherosclerotic plaque components.

Microbubble contrast agents are detectable because the frequencies used by the ultrasound imaging system will cause the expansion and contraction of the microbubbles. At high acoustic power, the oscillations of the microbubbles become asymmetric, and at a high enough acoustic power, the microbubbles may be induced to rupture. This behavior, known as nonlinear resonance, causes the microbubbles themselves to become intense ultrasonic emitters that produce unique acoustic signals based on their unique spectrum and profile. Clinically available ultrasound machines can distinguish nonlinear signals from tissue backscattering. The imaging system varies with the incident ultrasonic frequency, pulse waveform, sound power, whether the microbubbles are induced to burst, and the detection frequency. On two-dimensional images, all system configurations display processed microbubble signals with increased intensity. Micro air bubbles are used for the expansion of molecular imaging in the existing echocardiography applications. Their application by intravenous injection of microbubbles, and their rheological equivalent to that of red blood cells in blood vessels, creates a turbid blood pool to delimit the endocardial boundary and allows for a more accurate assessment of left ventricular systolic function. Two inflatable microsphere ultrasound contrast agents are currently in use for indications approved by the FDA.

### 4.4. Multimodal Imaging

Ultrasound is widely used, safe, and inexpensive, but has insufficient penetration for noninvasive imaging of deep blood vessels, including coronary arteries, with high spatial resolution or sensitivity. SPECT and PET have high sensitivity, but their spatial resolution is limited. Another disadvantage is the use of radioactive materials. In contrast, MRI is slightly less sensitive than SPECT and PET and requires a longer imaging time, but it is safe and has good resolution (~10 μm with a strong magnetic field). In contrast, CT has the advantages of a fast scan time and superior performance in coronary angiography at the cost of limited sensitivity, the use of nephrotoxic drugs, and ionizing radiation. Optical imaging techniques, such as near-infrared fluorescence reflection, or fluorescent molecular tomography, have excellent sensitivity and temporal resolution and allow for precise determination of the tissue distribution of the probe with an in vitro fluorescence microscope. However, until now, this technique has only been used noninvasively to monitor superficial structures due to the limited ability of light to penetrate tissue.

Because there is no single imaging method that ideally meets all research and medical needs, multiple imaging agents can be integrated into a unified platform. Multimodal nanoparticles with complementary imaging agents provide highly sensitive detection, anatomical localization, and data validation from different imaging modes. Nanoparticles combining optical, MR, and PET imaging agents have been reported. Imaging of macrophage-labeled multimodal nanoparticles by PET and MRI has been demonstrated. The availability of multimodal imaging devices that combine MRI, CT, optics, PET, and ultrasound facilitates simultaneous anatomical and molecular imaging. Functional PET imaging has been combined with CT to pinpoint radioisotope-labeled macrophages. Similarly, functional 19F MRI and fluorescence combined with anatomic MRI and CT, respectively, have been reported. Chen et al. [103] prepared a targeted nano-probe (called IMTP-Fe_3_O_4_-PFH NPs) with enhanced ultrasound (US), photoacoustic (PA), and magnetic resonance (MR) properties for direct and noninvasive visual imaging of ischemic myocardial models in rats. The probe has the characteristics of nanoscale, good stability, ADV, and safety. It has an obvious targeting effect on hypoxia-damaged cells and rat heart models. After NPs were injected into the tail vein of model rats, the in vivo imaging results showed that the US/PA/MR signal was significantly enhanced, indicating that this nano-probe is highly feasible for distinguishing ischemic myocardium. This makes up for the shortcomings of different imaging methods and provides a new idea for accurate diagnosis.

## 5. Conclusions and Perspectives

Myocardial injury can cause a strong inflammatory response, and inflammatory cells infiltrate into local tissues, activate myofibroblasts and vascular endothelial cells, promote tissue repair and scar formation, and at the same time lead to poor myocardial tissue fibrosis remodeling and aggravated myocardial cell apoptosis, leading to arrhythmia. There is increasing evidence that blocking the inflammatory signaling pathway can prevent adverse ventricular remodeling by promoting cardiac repair and extracellular matrix metabolism, which plays an important role in myocardial injury and repair. From this point of view, targeting anti-inflammatory therapy after myocardial injury has very important clinical significance. Future treatment strategies may shift in a new direction, ensuring the optimal time to regulate inflammation and prevent it from getting out of control. In addition, finding suitable inflammatory biomarkers to reflect the myocardial inflammatory state in patients with heart failure is the key to rationally regulating inflammation and preventing poor remodeling that could lead to heart failure in the future.

Other alternative research, such as implanting engineered cardiac muscle patches, formulating stem cell-carried injectable hydrogels, and developing novel analogs of endogenous cytokines by molecular bioengineering techniques, is ongoing. Significant advances in biotechnology, tissue engineering, biomimetics, and polymer science have enabled the development of nanomaterials with additional properties, such as controlled release, immune escape, and long blood circulation times, which has promoted the development of nanomaterials for medical applications. Thanks to these advantages, functional nanomedicine may become a new intersection between molecules, cells, tissues, biomaterials, and biomechanical engineering solutions.

However, there are still some obstacles and challenges to the development of nanomaterials. More research is required to illustrate the mechanisms by which these nanomaterials influence cell behavior. For example, as a tissue engineering material, the triggers for cell adhesion and cell response to hydrogel softness are still unclear. Also, we need to be aware that the size and synthesis of nanomaterials are not uniform, such as polymerized nanoparticles and nanozymes. As for nanoprobes, biological barriers to nanoprobe transport prevent bioimaging and biosensing at specific sites, such as some specific organelles in living cells or target tissues in vivo, so that their bioavailability at proper imaging and sensing outcomes are limited. As cardiac nanotechnology is still in its infancy, scientists should examine the experiences in the field of cancer nanotechnology, such as by applying optimal protocols to experimental setups and reporting the essential information (e.g., nanomaterial’s physicochemical properties, detailed information on the nano-bio interfaces, and biological identity of nanoparticles), which enables them to become a better and more comprehensive alternative to current cardiac injury and repair therapies in the future.

## Figures and Tables

**Figure 1 pharmaceutics-14-00758-f001:**
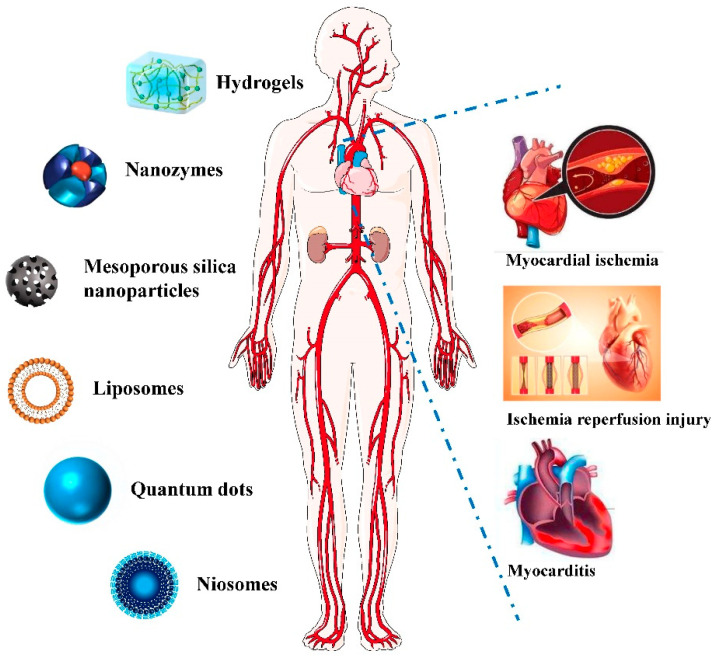
Application of Nano-theranostics for myocardial injury.

**Figure 2 pharmaceutics-14-00758-f002:**
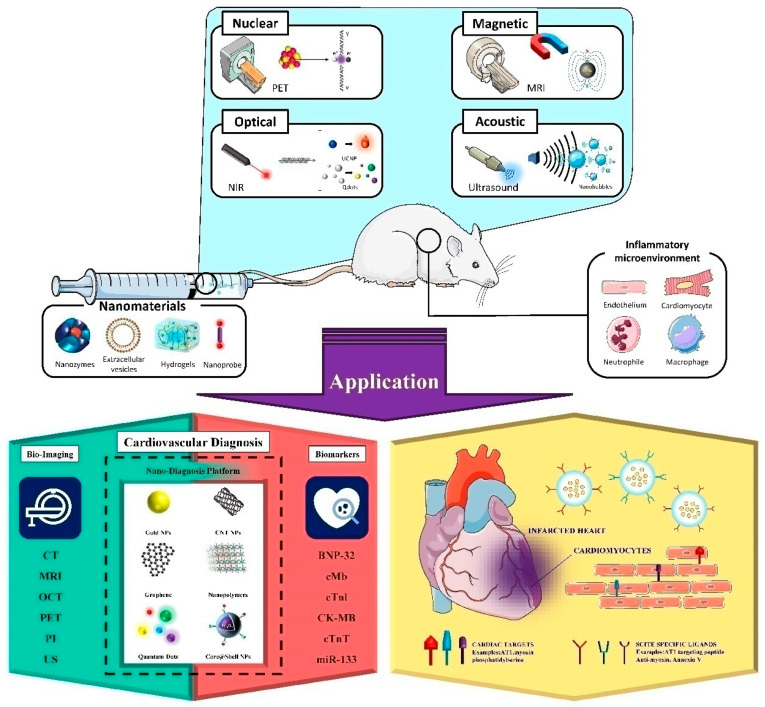
Application of nanotheranostics in the diagnosis and treatment of the myocardial inflammatory microenvironment.

**Table 1 pharmaceutics-14-00758-t001:** The application of functionalized nanomaterials in myocardial repair.

Nanocarrier	Size	Effective Constituent	Cargo Loading	Model	Type of Disease	Clinical Outcomes	Ref.
Nanofibrous gelling microspheres (NF-GMS)		60–90 μm	Human embryonic stem cell derived cardiomyocytes (hESC-CMs)	Coincubation	Myocardial infarction rat model	Myocardial infarction	achieve the highest reported engraftment of CMs to date, reduce infarct size, enhance integration of transplanted CMs, stimulate vascularization in the infarct zone, and leads to a substantial recovery of cardiac function	[36]
Triazole-(triazole-thiomorpholine dioxide [TMTD] alginate)	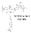	1.5 mm	Mesenchymal stem cells (MSCs)	Coincubation	Myocardial infarction rat model	Acute myocardial infarction	demonstrated in vivo therapeutic application of TMTD-alginate MSC capsules for improvement of ventricular functioning and remodelling in a post-MI rat model	[37]
Gold nanoparticle -hyaluronic acid (AuNP-HA) hydrogel	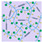	-	Human induced pluripotent stem cells cardiomyocytes (hiPS-CMs)	Coincubation	Myocardial infarction mouse model	Myocardial infarction	ameliorated the electrical conduction block of the ventricle, augment the angiogenic capacity, contribute to improved heart function and reduce ventricular remodeling after MI	[38]
Recombinant human ferritin nanocage (FTn)	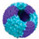	12 nm	Manganese (Mn) metal nanoparticle	In situ synthesis	cardiac ischemia–reperfusion mouse model	Cardiac ischemia–reperfusion	alleviate of mitochondrial oxidative injury and enhance the recovery of heart functionality	[42]
Nitric oxide synthase (NOS)-like nanoplatform (NanoNOS)	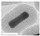	130 ± 2.3 nm	Noble metal nanoparticle	The modified seed-Mediated method	HUVEC (human umbilical vein endothelial cells) and THP-1 (human acute monocytic leukemia)	Cardiovascular diseases	enhanced the intracellular NO production, greatly diminished injury-induced monocyte-endothelial cell adhesion and help prevent cardiovascular disease	[43]
Cardiomyocyte (CM)-derived CD172aþ EVs	-	0.1–0.5 μm	CM-derived CD172aþ EVs	-	Hypoxic human-induced pluripotent stem cell-derived cardiomyocytes	Cardiovascular diseases	represent a new class of biomarker for myocardial diseases, especially aortic stenosis	[49]
Cardiac progenitor cells-derived exosomes (CPCs-Ex)	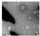	30–100 nm	CPCs-Ex	-	Viral myocarditis rat modal	Viral myocarditis	attenuate cardiomyocyte apoptosis, repair the cardiomyocyte function	[50]
Primary cardiomyocyte-conjugated and 17β-estradiol-loaded perfluorocarbon nanoprobes (PCM-E2/PFPs)	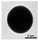	418 ± 11 nm	Primary cardiomyocyte and 17β-estradiol	Click chemistry	Cardiac hypertrophy rat model	Cardiac hypertrophy	promises to be a potential clinical tool for off-target therapeutics delivery as well as ultrasound contrast Enhancers for theranostics on myocardial pathophysiology	[54]
Lanthanide metal-organic framework nanoprobes (Eu−QPTCA)	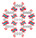	150–250 nm	Europium (Eu) metal nanoparticle	Reaction	-	Acute myocardial infarction	Show superior selectivity and reliability toward the practical detection of creatine kinase activity in human serum, indicating the great significance in the early diagnosis of acute myocardial infarction	[56]
Reactive oxygen species -scavenging material nanoparticles (TPCD NP)	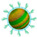	101 nm	Reactive oxygen species-scavenging material	Nanoprecipitation/self assembly method	DOX-induced heart failure mouse model	Heart failure	Efficaciously ameliorate DOX-induced heart failure largely by site-specific attenuation of oxidative stress in the heart	[57]
*Ganoderma lucidum spore oil* (GLSO) @P188/PEG400 nanosystem	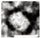	90 nm	*Ganoderma lucidum spore oil* (GLSO)	Homogenization method	X-ray irradiation mouse	Radiation-induced heart disease	Shield the heart from X-rays in vivo, as evidenced by attenuating cardiac dysfunction and fibrosis, accompanied by significant alleviation of X-ray-induced necrosis	[58]
Poly (lactic acid/glycolic acid) nanoparticle	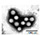	223 nm	Pioglitazone	-	Mouse and porcine myocardial IR injury model and MI model	Myocardial infarctionandCardiac ischemia–reperfusion	Protected the heart from IR injury and cardiac remodeling by antagonizingmonocyte/macrophage-mediated acute inflammation and promoting cardiac healing afterAMI	[59]
Macrophagemembrane coated nanoparticles (MMNPs)	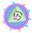	-	microRNA199a-3p	Extrusion	Myocardial infarction mouse model	Myocardial infarction	Ameliorate left ventricular remodelingand cardiac functions, and protect against MI	[60]
Melanin nanoparticles (MNPs)/alginate (Alg) hydrogels	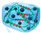	-	Melanin nanoparticles	Divalent cations (Ca^2+^) cross-linking method	Myocardial infarction rat model	Myocardial infarction	Regulate ROS and the immune MI microenvironment for cardiac repair	[61]
MIONs loaded NPs (PP/PS@MIONs)	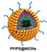	50–80 nm	PP/PS@MIONs	The thin-film dispersion method	Myocardial infarction rat model	Myocardial infarction	Show good inflammation-resolving effects and imaging ability, significantly improve the treatment efficacy of MI at an early stage	[62]
Human apolipoprotein A-I nanoparticles (n-apoA-I)	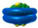	-	apoA-I	-	Preclinical mouse model of myocardial infarction	Myocardial infarction	Reduce the systemic and cardiac inflammatory response through direct actions on both the ischemic myocardium and leukocytes	[63]
Liposomes	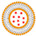	100 nm	Methotrexate	-	Lipopolysaccharide-injury skeletal muscle mouse model and myocardial ischemia reperfusion injury mouse model	Chronic inflammatory diseases	Demonstrated that the drug-loaded liposomes would be released when neutrophils migrate to the inflamed tissue	[64]
FH peptide-modified neutrophil-mimicking membranes on mesoporous silicon nanoparticles (FNLM-miR)	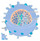	170 nm	mRNAs	Coextrusion	Cardiac ischemia–reperfusion mouse model	Cardiac ischemia–reperfusion	Induce cardiac reprogramming efficiently, leading to improved cardiac function and mitigated fibrosis after myocardial I/R injury	[65]
Prostaglandin E2-platelet-inspired nanocell (PEG_2-_PINC)	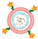	195 nm	Cardiac stem/stromal cells (CSCs)	Double emulsion method	Cardiac ischemia–reperfusion mouse model	Cardiac ischemia–reperfusion	Can achieve targeted delivery of therapeutic payloads to the injured heart, augment cardiac function and mitigate heart remodeling	[66]

## Data Availability

Not applicable.

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
