# Peer review of "Designer Functional Nanomedicine for Myocardial Repair by Regulating the Inflammatory Microenvironment"

_pharmaceutics, 2022, doi:10.3390/pharmaceutics14040758_

Round 1

Reviewer 1 Report

The manuscript entitled "Designer functional nanomedicine for myocardial repair by 2 regulating the inflammatory microenvironment "  describe various nanoparticles efficiency in myocard tissue repair after ischemic and non-ischemic damage. The description is very interesting and the objective is generous.

Still, the following observation has to be made:

Is their review a systematic review or a descriptive (narrative review). This aspect should be mentioned in the manuscript in a separate section. In the meaning time they need to describe the method they use for this manuscript. 

Author Response

Response to Reviewer #1

Reviewer1: comments and Suggestions for Authors

The manuscript entitled "Designer functional nanomedicine for myocardial repair by regulating the inflammatory microenvironment " describe various nanoparticles efficiency in myocardial tissue repair after ischemic and non-ischemic damage. The description is very interesting and the objective is generous.

Response: The reviewer's positive comments are greatly appreciated. This is an integrative review, which critically summarizes representative advances in functional nanomedicines targeting the inflammatory microenvironment for myocardial repair. Through the integration of nanomedicine design strategies to provide an overview of the researchers in this field. Finally, the prospects and challenges in this field are discussed in order to raise people's attention.

Still, the following observation has to be made:

Is their review a systematic review or a descriptive (narrative review). This aspect should be mentioned in the manuscript in a separate section. In the meaning time they need to describe the method they use for this manuscript.

Response: Thanks to the reviewer for his/her professional advice, which is very helpful in clarifying the form and positioning of our manuscript. This manuscript is a descriptive review which systematically reviews the application of nanomaterials in myocardial repair in recent years based on the inflammatory microenvironment.

Reviewer 2 Report

The review by Liu et al is an interesting collection of research experience in the field of functional nanomedicine for myocardial repair. According to me it is of very interest for the scientific community in this specific field as well as in other nanomedicines sub fields not related to myocardial injury.

Please find enclosed few comments I hope to be useful for ameliorate the work. 

  1. Please revise typing errors in the abstract:

Page 1, line 17. add a space between the word “infraction” and “is”

2- Reference about causes of myocardial injury (Ref 2) should be replaced/integrated with another, much more specific review, defining and describing in a direct way the causes of AMI. The same for reference 9 and 10 (review 9 is on extracellular vesicles, ref 10 is focused mainly on prostatitis)

3- page 2 line 70-71 “Currently, nearly 25% of new drugs under development and in clinical trials are related 70 to cardiovascular diseases” a reference for this statistic should be added

4- The sentence “With the rapid development of materials science, nanotechnology and biomedical engineering, a new research field, nanomedicine, in which nanoparticles are used in the 73 prevention, treatment and imaging of diseases, has produced a series of fruitful applications.” poorly readable. Please rephrase.

5- page 2 line 83. What is intended as Functional Nanoparticles?

6- Page 3 line 113-127. This part should be moved in a section describing the genesis of the inflammatory myocardial injuries, and replaced by treatment options as stated in the title. Otherwise the “treatment“ should be removed from the subtitle2.1.

7- All the information about the phisiopatology of miocardial inflammatory deseases should be condensed in one paragraph. The section should highlight myocardial cellular population and microenvironment, metabolism and their change i pathological conditions, to give to the reader a clear shenario of the nanomedicines targets and  possible barriers to the delivery. See for example Page 3, lines 143-146, Page 6, lines 276-282, Page 12, 343-359

8-all the cons of using nanoparticles should be concentrated. Repetitive parts are present. For example, pag 7 lines 283-285 and 292-295.

9- page 7 line 319. Please replace “drug aggregation” with “drug accumulation”

10- page 8 336-338. Please rewrite avoiding “drug-carrying nano carrier”

11- Page 12 line 363-364. the concept of hierarchical targeted delivery system should be explained

12- Page 4, line 198. the presence of a recombinant human ferritin nanocage and coverage with TPP, make of Yue's nanoparticles a hybrid nanozyme and not a nanase which for definition is inorganic nanoparticle with catalitic activity

13- page 5, lines 245-246. The definition of nanoprobe could not be appropriate [https://doi.org/10.1016/j.nantod.2011.02.007]. The scope of nanoprobe is primarily the targeted imaging of biochemical changes in living systems. A combinatorial theranostic approach is an interesting evolution of the concept, where drug delivery is integrated with diagnosis and therapeutic monitoring.

Author Response

Response to Reviewer #2

The review by Liu et al is an interesting collection of research experience in the field of functional nanomedicine for myocardial repair. According to me it is of very interest for the scientific community in this specific field as well as in other nanomedicines sub fields not related to myocardial injury.

Please find enclosed few comments I hope to be useful for ameliorate the work.

  1. Please revise typing errors in the abstract:

Page 1, line 17. add a space between the word “infraction” and “is”

Response:Thank you very much for pointing out the problem. We have add a space between the word “infraction” and “is”,and makered in red.(line17)

2- Reference about causes of myocardial injury (Ref 2) should be replaced/integrated with another, much more specific review, defining and describing in a direct way the causes of AMI. The same for reference 9 and 10 (review 9 is on extracellular vesicles, ref 10 is focused mainly on prostatitis)

Response:Thank you for your comments. The reference has been replaced with another one which is much more specific review, ,and makered in red.(line1-4, Page 21)

3- page 2 line 70-71 “Currently, nearly 25% of new drugs under development and in clinical trials are related 70 to cardiovascular diseases” a reference for this statistic should be added

Response:Thank you very much for your suggestions. According to your comment, we have had added the reference ,and makered in red.(line70-71, Page2)

4- The sentence “With the rapid development of materials science, nanotechnology and biomedical engineering, a new research field, nanomedicine, in which nanoparticles are used in the 73 prevention, treatment and imaging of diseases, has produced a series of fruitful applications.” poorly readable. Please rephrase.

Response:Thank you for pointing out the problem. We have rephrased the sentence,and makered in red.( line73-76, Page2)

5- page 2 line 83. What is intended as Functional Nanoparticles?

Response:Thank you for your comments. We have redefined the means of functional nanomaterials, ,and makered in red.( line76-78, Page2)

6- Page 3 line 113-127. This part should be moved in a section describing the genesis of the inflammatory myocardial injuries, and replaced by treatment options as stated in the title. Otherwise the “treatment“ should be removed from the subtitle2.1.

Response:Thank you for your comments. We have modified this part ,and makered in red.(line105)

7- All the information about the phisiopatology of miocardial inflammatory deseases should be condensed in one paragraph. The section should highlight myocardial cellular population and microenvironment, metabolism and their change i pathological conditions, to give to the reader a clear shenario of the nanomedicines targets and possible barriers to the delivery. See for example Page 3, lines 143-146, Page 6, lines 276-282, Page 12, 343-359

Response:Thank you very much for your comments. We have modified this part ,and makered in red.(line112-113, line143, Page 3)

8-all the cons of using nanoparticles should be concentrated. Repetitive parts are present. For example, pag 7 lines 283-285 and 292-295.

Response:Thank you very much for your comments. We have concentrated all the cons of using nanoparticles and delete the repetitive parts,and makered in red.(line675-689, Page20)

9- page 7 line 319. Please replace “drug aggregation” with “drug accumulation”

Response:Thank you very much . We have replaced “drug aggregation” with “drug accumulation”, and makered in red.( line331, Page8)

10- page 8 336-338. Please rewrite avoiding “drug-carrying nano carrier”

Response:Thank you for pointing out the problem. We have deleted it throughout the text.

11- Page 12 line 363-364. the concept of hierarchical targeted delivery system should be explained

Response:Thank you for comments. The concept of hierarchical targeted delivery system have been explained,and makered in red.( line375-378, Page13)

12- Page 4, line 198. the presence of a recombinant human ferritin nanocage and coverage with TPP, make of Yue's nanoparticles a hybrid nanozyme and not a nanase which for definition is inorganic nanoparticle with catalitic activity

Response:Thank you for your valuable comments.We have redefined the concept of nanase.

13- page 5, lines 245-246. The definition of nanoprobe could not be appropriate [https://doi.org/10.1016/j.nantod.2011.02.007]. The scope of nanoprobe is primarily the targeted imaging of biochemical changes in living systems. A combinatorial theranostic approach is an interesting evolution of the concept, where drug delivery is integrated with diagnosis and therapeutic monitoring.

Response:Thank you for your valuable comments.We have redefined the concept of nanoprobe, and makered in red. ( line243-247, Page5)

Reviewer 3 Report

Dear authors,

I have read the review, 'Designer functional nanomedicine for myocardial repair by regulating the inflammatory microenvironment' with high interest. The review is well written and covered the recent literature. I have some minor comments before acceptance.

  1. What are the challenges these nanomedicines? This section is not covered, challenges can be included in detail including but not limited to production, cost, reproducibility, targeting, toxicity and degradability. What are the regulatory challenges?
  2. The conclusion is very generic and needs more authors and futuristic perspectives. When do these trials may be clinically approved? What other alternative research is currently ongoing apart from these nanomedicine strategies including 3D printed device or others. Please mention.
  3. Some recent references are missing, eg: a) Biomaterials science 8, no. 18 (2020): 5061-5070. b) Cardiovasc Res. 2019 Feb 1;115(2):419-431. c) Bioeng Transl Med. 2020 Nov 19;6(2):e10197. 

Author Response

Response to Reviewer #3

I have read the review, 'Designer functional nanomedicine for myocardial repair by regulating the inflammatory microenvironment' with high interest. The review is well written and covered the recent literature. I have some minor comments before acceptance.

  1. What are the challenges these nanomedicines? This section is not covered, challenges can be included in detail including but not limited to production, cost, reproducibility, targeting, toxicity and degradability. What are the regulatory challenges?

Response: Thank you very much for your suggestions. According to your comment, we have had the challenges covered including production, cost, reproducibility, targeting, toxicity and degradability and marked in red(line678-692).

  1. The conclusion is very generic and needs more authors and futuristic perspectives. When do these trials may be clinically approved? What other alternative research is currently ongoing apart from these nanomedicine strategies including 3D printed device or others. Please mention.

Response: Thank you for your valuable comments. According to your comments, we have had supplemented the conclusion and marked in red(line669-671、675-677).

  1. Some recent references are missing, eg: a) Biomaterials science8, no. 18 (2020): 5061-5070. b) Cardiovasc Res. 2019 Feb 1;115(2):419-431. c) Bioeng Transl Med. 2020 Nov 19;6(2):e10197. 

Response: Thank you for your comments. The relevant literature has been supplemented in the manuscript and marked in red(line175-178、323-330).

Reviewer 4 Report

The current review presents an exciting and timely review of literature relating to the use of nanomedicine in combating myocarditis and cardiac inflammation. The review is nicely structured and presented, with the use of informative figures and tables that is well supplemented with discussion. Over 100 key pieces of literature, from both contemporary and older sources are cited. I think the manuscript warrants acceptance in Pharmaceutics, but could be improved by including a limitations section that critically reviews the potential of nanomedicine in treating myocarditis at the completion of the review manuscript. 

Author Response

Response to Reviewer #4

Reviewer 4:Comments and Suggestions for Authors

The current review presents an exciting and timely review of literature relating to the use of nanomedicine in combating myocarditis and cardiac inflammation. The review is nicely structured and presented, with the use of informative figures and tables that is well supplemented with discussion. Over 100 key pieces of literature, from both contemporary and older sources are cited. I think the manuscript warrants acceptance in Pharmaceutics, but could be improved by including a limitations section that critically reviews the potential of nanomedicine in treating myocarditis at the completion of the review manuscript.

Response: Thanks to the reviewers for your positive comments on our manuscript, we have critically discussed the potential application of nanomaterials in myocarditis as suggested and marked in red in Section 5, Conclusions and Prospects of the manuscript.